# *Bacteroides fragilis* Promotes Mesenchymal Subtype in Colorectal Cancer

**DOI:** 10.3390/cancers17233822

**Published:** 2025-11-28

**Authors:** Shin Young Chang, Jihye Park, Soo Jung Park, Jae Jun Park, Jae Hee Cheon, Dong Keon Kim, Tae Il Kim

**Affiliations:** 1Division of Gastroenterology, Department of Internal Medicine, Institute of Gastroenterology, Severance Hospital, Yonsei University College of Medicine, 50-1 Yonsei-ro, Seodaemun-gu, Seoul 03722, Republic of Korea; 2Graduate School of Medical Science, Brain Korea 21 Project, Yonsei University College of Medicine, Seoul 03722, Republic of Korea; 3Yonsei Cancer Prevention Center, Severance Hospital, Yonsei University College of Medicine, Seoul 03722, Republic of Korea

**Keywords:** CMS4, microbiome, patient-derived organoids, *Bacteroides fragilis*, colorectal cancer

## Abstract

Colorectal cancer (CRC) is a heterogeneous disease with an aggressive stromal subtype, CMS4, that relies on the tumor microenvironment (TME) for progression and malignancy. The microbiome is a key component of the TME; however, its role in determining CRC subtypes remains unclear. We identified enterotoxigenic *Bacteroides fragilis* as a microbial species enriched in CMS4 tumors and demonstrated its potential to promote CMS4-associated transcriptomic signatures in CMS2 tumors. These findings suggest that targeting tumor–microbiome interactions may offer a novel strategy for treating stromal-driven, invasive CRC subtypes.

## 1. Introduction

Colorectal cancer (CRC) is a molecularly heterogeneous disease. The Consensus Molecular Subtype (CMS) classification system established by the CRC Subtyping Consortium in 2015 defines four major subtypes based on transcriptomic profiles—CMS1 (microsatellite instability [MSI]-immune), CMS2 (canonical), CMS3 (metabolic), and CMS4 (mesenchymal)—which together encompass approximately 87% of all CRC cases [1,2]. CMS4 is characterized by stromal infiltration, elevated transforming growth factor β signaling, angiogenesis, and epithelial–mesenchymal transition, and is associated with poor prognoses and a high potential for metastasis [3,4]. Although CMS4 is shaped largely by tumor–stroma interactions, the causative factors driving its emergence remain poorly defined. The tumor microenvironment (TME), which is composed of cancer-associated fibroblasts, immune cells, the extracellular matrix (ECM), and vasculature, plays a pivotal role in tumor progression and subtype determination [5,6,7,8].

Gut microbiota—comprising primarily species of *Firmicutes*, *Bacteroidetes*, *Actinobacteria*, and *Proteobacteria*—maintain intestinal homeostasis but can contribute to tumorigenesis when dysregulated [9,10,11,12,13,14,15]. Certain microbial species, such as *Fusobacterium nucleatum*, pks-positive *Escherichia coli*, and enterotoxigenic *Bacteroides fragilis* (ETBF), have been implicated in CRC initiation and progression through mechanisms involving chronic inflammation, DNA damage, and modulation of host signaling pathways [16,17]. While non-toxigenic *B. fragilis* (NTBF) is a commensal symbiont that can exhibit anti-inflammatory properties and contribute to immune homeostasis [18,19,20], the enterotoxigenic strain (ETBF) secretes *B. fragilis* toxin (BFT), which has been implicated in pathological processes. In this study, we specifically focus on the role of ETBF in the context of aggressive CRC subtypes. ETBF has been shown to induce tumorigenesis via the Wnt/β-catenin pathway and STAT3/Th17 axis, and to promote cancer stemness through NANOG and SOX2 factors [21,22,23,24,25].

Recent studies suggest that microbial compositions may differ by CRC subtype, with specific bacterial species enriched in CMS1, CMS2 and CMS3 tumors [26]. However, the landscape of the CMS4 microbiome remains largely unexplored, and its potential role in subtype induction remains unclear.

For this study, we hypothesized that the tumor-associated microbiome acts as a key modulator of both the TME and the molecular subtype of CRC. To test the hypothesis, we compared the microbial composition of mesenchymal (CMS4) and epithelial (CMS2, CMS3) CRC subtypes by analyzing 16S rRNA sequencing data from 25 CRC tissue samples. This is the first study to identify microbial taxa specifically enriched in CMS4 and demonstrate the ability of key bacterial species to induce CMS4-like phenotypes using patient-derived organoids (PDOs) and a co-culture model. Our findings reveal a novel functional role of the microbiome in shaping aggressive CRC subtypes.

## 2. Materials and Methods

### 2.1. Patient Cohort and Samples

Tissue samples of CRC tumors were collected from 25 patients (Table 1). Written informed consent was obtained from patients prior to the beginning of the project. The study was carried out in accordance with the Declaration of Helsinki and with the approval of the Institutional Review Board of Yonsei University College of Medicine, Severance Hospital (IRB No. 4-2012-0859), and all experiments were performed according to the appropriate guidelines and regulations. The patients ranged in age between 26 and 92 years (mean, 64), with 16 male patients and 9 female patients. Biopsies were obtained via surgery or colonoscopy, and the tumors samples included 9 from the right colon, 7 from the left colon and 9 from the rectum. Histologically, 8 tumors were well-differentiated, 15 were moderately differentiated, and 2 were poorly differentiated. Table 1 presents detailed characteristics of the patient cohort.

### 2.2. Cell Culture and Reagents

Cells of the human myofibroblast cell line (CCD-18Co, RRID:CVCL_1129) and monocytic myeloid cell line (THP-1, RRID:CVCL_0006) were purchased from the Korean Cell Line Bank (Seoul, Republic of Korea). All cell lines were routinely tested for mycoplasma contamination (PCR-based assay, negative results) and authenticated by short tandem repeat (STR) profiling prior to use. Experiments were conducted only with cells that passed both quality-control checks. All cells were cultured in high-glucose HyClone^TM^ Dulbecco’s Modified Eagle’s Medium (DMEM) (Cytiva, Marlborough, MA, USA) supplemented with 10% fetal bovine serum (FBS) (Gibco-Life Technologies, Waltham, MA, USA), and 100 units/mL penicillin and 100 mg/mL streptomycin (P/S) (Invitrogen, Waltham, MA, USA). All cells were maintained in a 5% CO_2_ incubator at 37 °C.

For subcultures, cells were seeded at a density of 1 × 10^6^ cells/mL in 100 mm culture plates and cultured until they reached 80% confluency. Cells were washed using 1× phosphate-buffered saline (PBS) (GeneTech, Asan, Chungcheong, Republic of Korea). Adherent cells were detached from the plate surface using 0.25% trypsin-EDTA (Invitrogen) and neutralized with high-glucose DMEM containing 10% FBS and 1% P/S. Cell density was measured using a cell counter (NanoEnTek, Seoul, Republic of Korea) and trypan blue (Gibco-Life Technologies).

For each co-culture experiment, 1  ×  10^5^ cells of each cell line (THP-1 and CCD-18Co) were mixed and cultured in the lower well of a 12-well Transwell plate (Costar^®^ Corning Incorporated, Corning, NY, USA).

### 2.3. Patient-Derived Tumor Organoid Culture

Tissue samples from the patient cohort were used to establish PDOs (Table 1). Cells were dissociated from tissue samples and embedded in growth factor-reduced Matrigel (Corning Inc.) and seeded in 48-well plates (20 μL of Matrigel per well). The Matrigel was left to polymerize at 37 °C for 15 min, and a 250 μL/well basal culture medium of advanced DMEM/F12 medium supplemented with 1% P/S, 1X Glutamax, 1X N2, B27 (Invitrogen), and 1 mM N-acetyl-l-cysteine (Sigma-Aldrich, St. Louis, MO, USA) was added with the following optimized growth-factor combinations: 2  mM l-glutamine, 10  mM nicotinamide, 10  nM gastrin I, 500  nM A-83-01, 10  μM SB202190, 50  ng/mL EGF, 100 ng/mL noggin, and 100 ng/mL Wnt3a.

All organoids were maintained in a 5% CO_2_ incubator at 37 °C. For passaging, PDOs were washed with PBS and dissociated using 1× TrypLE (Gibco-Life Technologies), after which they were seeded in 20 µL Matrigel and overlaid with PDO growth media. For monoculture experiments, 6 × 10^3^ cells were seeded in a 24-well culture plate and grown for 7 days in a PDO growth media. For the co-culture experiments, 3 × 10^3^ organoid cells were grown in the upper chambers of the Transwell plates for 7 days. PDO cells in each condition were measured across a week of growth.

### 2.4. Bacterial Culture and Treatment

The ETBF strain (isolated from colon cancer patients by the Department of Laboratory Medicine, Yonsei University Health System) was grown as suspension cultures with 5% sheep’s blood (MBcell, Seoul, Republic of Korea), Wilkins Chalgren (Oxoid, Basingstoke, Hampshire, UK), and 1 g/L sodium thioglycolate (Sigma-Aldrich) in anaerobic conditions using a 2.5-L anaerobe jar (Oxoid) and CO_2_ GasPak (Becton Dickinson, Franklin Lakes, NJ, USA). The identity of the bacterial isolates was confirmed by polymerase chain reaction (PCR) amplification with *bft*-specific primers, followed by agarose gel electrophoresis. (Appendix A). Bacterial cell density was calculated using an enzyme-linked immunosorbent assay plate reader (Molecular Devices, San Jose, CA, USA) and an optical density at a wavelength of 600 nm. Cell lines and PDOs were treated with bacteria in a 1:100 cell-to-bacteria ratio. Cell lines in serum/antibiotic-free media were treated with bacteria for 2 h in anaerobic conditions. PDOs suspended in serum/antibiotic-free media were treated with bacteria for 1.5 h in anaerobic conditions. After bacteria treatment, growth media supplemented with Normocure, a broad-spectrum antibiotic (100 µg/mL, Invitrogen), was added to the cells/PDO and cultured in aerobic conditions.

CellTracker red (CMTPX; Invitrogen) was used to label the bacteria to confirm cell infection. The bacteria were treated with 1 µM of product and stored in anaerobic conditions for 15 min, after which they were removed and washed with PBS before being used for cell line or PDO treatment. Bacterial attachment to PDOs after being labeled with the fluorescence tracker was confirmed using a fluorescence microscope (Olympus Corporation, Tokyo, Japan) (Appendix A).

### 2.5. Genomic DNA Isolation, Gene Library Amplification and Gel Electrophoresis

Bacterial genomic DNA was isolated using a boiling lysis method with minor modifications. A 1 mL bacterial suspension was transferred to a 1.5-mL microcentrifuge tube and centrifuged at 13,000 rpm for 2 min. The supernatant was discarded, and the pellet was washed with 1 mL of PBS by centrifugation at 13,000 rpm for 1 min. The pellet was then resuspended in 100 µL of distilled water and boiled at 100 °C for 5 min. Cell debris was removed by centrifugation at 5000 rpm for 10 min, and the supernatant containing genomic DNA was transferred to a new 1.5 mL microcentrifuge tube and stored at −20 °C until use.

Genomic DNA was subjected to PCR amplification using a *bft*-specific primer set, as described by Odamaki et al. [27,28]. The set targets and efficiently amplifies all three known *bft* isotypes. Amplification was performed under cycling conditions outlined by Odamaki et al. using DreamTaq DNA polymerage (Thermo Fisher Scientific, Waltham, MA, USA): initial denaturation at 94 °C for 3 min; 35 cycles of denaturation at 94 °C for 30 s, annealing at 66 °C for 30 s, and extension at 72 °C for 30 s; followed by a final extension at 72 °C for 2 min. The expected 296 base-pair amplicon was separated on a 1.5% (*w*/*v*) agarose gel prepared in 1× TAE buffer and stained with Redsafe (Intron Biotechology, Seongnam, Gyeonggi, Republic of Korea). Electrophoresis was conducted at 125 V for approximately 30 min, and DNA bands were observed using a gel documentation imaging system, MiniBis Pro (DNr Bio-Imaging Systems, Jerusalem, Israel) (Appendix A).

### 2.6. RNA Isolation, Sequencing and Consensus Molecular Subtype Analysis

Total RNA was isolated using Trizol reagent (Gibco-Life Technologies) and RNeasy Plus Mini Kit (Qiagen, Hilden, NW, Germany) according to the manufacturer’s protocol. RNA quantity was assessed using a NanoDrop 2000 spectrophotometer (Thermo Fisher Scientific). Library preparation and sequencing were performed by a commercial service provider (Macrogen Incorporated, Seoul, Republic of Korea).

For CMS analysis, read counts from RNA sequencing data were processed in the R package CMScaller [29,30]. Based on the gene expression profiles of each sample, the CMS of CRC was predicted and classified.

### 2.7. 16S rRNA Gene Amplicon Sequencing

The entire experimental workflow, from DNA quality control to sequencing, was professionally conducted by LAS Incorporated (Seoul, Republic of Korea). Initially, the concentration of genomic DNA from 25 tissue samples taken from human colon tumors was quantified using Trinean Dropsense 96 spectrophotometer (Surplus Solutions, Woonsocket, RI, USA), and DNA integrity was verified via gel electrophoresis. Amplicon libraries were prepared following the Illumina 16S Metagenomic Sequencing Library Preparation protocol (Illumina Incorporated, San Diego, CA, USA). Specifically, the hypervariable V3-V4 regions of the 16S rRNA gene were amplified using locus-specific primers containing Illumina overhang adapter sequences. Following purification, a second PCR was performed to attach dual indices and sequencing adapters.

The quality and size distribution of the final libraries were validated using a dsDNA 910 Reagent Kit (Advanced Analytical Technologies, Ankeny, IA, USA). The validated libraries were then normalized, pooled in equimolar concentrations, and sequenced on the Illumina MiSeq platform. Sequencing was performed using a MiSeq Reagent Kit v3 (600-cycle) by Illumina to generate paired-end (2 × 300 base pair) reads.

### 2.8. Comparative Analysis of Gut Microbiota in Colorectal Cancer CMSs

To compare the microbial community across different subtypes of CRC, we analyzed 16s rRNA gene sequencing data using the EzBioCloud platform (CJ Biosciences, Seoul, Republic of Korea) [31]. CMS4 16s rRNA samples were used as one microbiome taxonomic profiling (MTP) set, and CMS2 and CMS3 as another MTP set for all microbiome analyses. Based on previous findings suggesting distinct biological pathways, samples were stratified into two groups for comparative analysis: the mesenchymal subtype (CMS4) and a combined epithelial-like subtype group (CMS2/3).

To assess intra-group (alpha) microbial diversity, we calculated the Chao1 index for richness and the Simpson index for evenness and dominance. To evaluate inter-group dissimilarities (beta diversity), we performed principal coordinate analysis (PCoA) on weighted UniFrac distance matrices, with statistical significance determined by 999 permutations. Finally, to identify specific taxonomic biomarkers that were differentially abundant between the CMS4 and CMS2/3 groups, we employed linear discriminant analysis (LDA) effect size (LEfSe).

### 2.9. Analysis of TCGA Validation Cohort

For a validation analysis using The Cancer Genome Atlas Colon Adenocarcinoma (TCGA-COAD) cohort (*n* = 444), we downloaded RNA-seq data using the TCGAbiolink (ver. 2.53.3) package in R to classify each sample into its subtype. Microbiome data for the same cohorts were obtained from the Bacteria In Cancer database (http://bic.jhlab.tw (accessed on 7 October 2024)). These two datasets were integrated to perform a random forest (ver. 4.7-1.2) analysis, identifying key microbial taxa that contribute to the prediction and classification of the CMS4 subtype.

### 2.10. Gene Set and Pathway Analysis

Gene set variation analysis (GSVA) was performed using the GSVA R package (ver. 1.52.3) from Bioconductor. The analysis was performed on normalized logCPM values from the RNA-seq data using the CMS4-specific set of 236 genes obtained from CMScaller. A Gaussian kernel cumulative distribution function was applied during the analysis. To elucidate the functional roles of differentially expressed genes, Kyoto Encyclopedia of Genes and Genomes (KEGG) pathway-enrichment analysis was conducted using the clusterProfiler package (ver. 4.12.6) in R. Additionally, preranked gene set enrichment analysis (GSEA) was performed using the clusterProfiler package (ver. 4.12.6). A custom ‘CANCER_MIGRATION_SUPERSET’ was generated by consolidating metastasis- and invasion-related gene sets (H, C2, C6) from the Molecular Signatures Database (MSigDB) based on normalized logCPM values.

### 2.11. Statistical Analysis

Statistical analysis used the free statistics software Jamovi (2.4) [32] and R (4.4.3) with a *p*-value < 0.05 established as the threshold for statistical significance. The specific test applied was determined by the experimental design. For comparisons between two groups, the Wilcoxon rank-sum test was utilized. For comparisons involving more than two groups, the Kruskal–Wallis test was performed, followed by Dunn’s post hoc test for pairwise analysis. *p*-values derived from multiple comparisons, such as in KEGG/GSVA analyses, were adjusted using the Benjamini–Hochberg method. To assess similarities between tissues and PDO co-cultured samples, a correlation matrix heatmap was drawn using Spearman’s correlation coefficient after normalization of RNA expression data with the patient-tissue data. Unless otherwise noted, all data are presented as mean ± SEM. Sample sizes (n), which reflect the number of independent biological replicates, are specified in the relevant figure legends

## 3. Results

### 3.1. Bacteroides fragilis Is Significantly Enriched in the Microbiome of CMS4 Tumor Tissues

To characterize the molecular and microbial landscapes of CRC, we performed whole transcriptome and 16S rRNA sequencing on tumor tissues from 25 patients with colon cancer (Figure 1A). Subsequently, we classified the CMS status of these 25 CRC patient tumor tissues: 36% of the samples were CMS2, 32% were CMS4, 16% were CMS3, and 8% were CMS1. The remaining 8% of the samples were unclassified (Figure 1B). CMS2 and CMS4 were the predominant subtypes, collectively representing more than two-thirds of the analyzed cases of CRC.

To investigate CMS-specific microbial profiles, we analyzed 16S rRNA sequencing data matched to tumor tissues from CRC patients classified as CMS2, CMS3, or CMS4. An alpha diversity analysis revealed that the CMS4 group exhibited significantly higher microbial diversity (Simpson index, *p* = 0.012) (Figure 1C), although species richness (Chao1 index) did not differ significantly between groups (*p* = 0.519) (Appendix A). A beta diversity analysis using PCoA revealed a significant difference in microbial community composition between CMS4 and CMS2/CMS3 samples (PERMANOVA F = 2.077, *p* = 0.017) (Figure 1D). LEfSe analysis further confirmed *B. fragilis* was the most significantly enriched species in CMS4 tissues, with an LDA score of 4.7 (*p* = 0.033), indicating a strong power to discriminate between CMS4 and CMS2/3 CRC (Figure 1E). Taxonomic profiling revealed enrichment of *B. fragilis* in CMS4 tissues, accounting for 14.04% of the bacterial community, compared with just 3.95% in CMS2/CMS3 tissues (Appendix A). *Fusobacterium nucleatum* and *Gemella morbillorum* were also more abundant in CMS4 (12.69%) than in CMS2/3 (4.28%) (Appendix A).

### 3.2. Analysis of the TCGA-COAD Dataset Reveals a CMS4-Specific Microbial Signature in Colorectal Cancer, Including a Notable Enrichment of Bacteroides

To further validate and expand on our findings in a larger, independent dataset, we analyzed the microbiome features within the CMS using RNA-seq data from a total of 444 TCGA-COAD samples. Classification of subtypes based on this RNA-seq data revealed that CMS4 was the most prevalent subtype, accounting for 29.7%, followed by CMS2 (27.5%), CMS1 (17.3%), and CMS3 (16.0%). The remaining 9.5% of the samples were classified as not applicable (Figure 2A).

To assess gut microbial community diversity in this cohort, samples were regrouped into CMS2/3 (merging CMS2 and CMS3) and CMS4 groups. Alpha diversity as measured by the Simpson index (richness and evenness within samples) showed no statistically significant difference between these two groups (Wilcoxon rank-sum test, *p* = 0.107) (Figure 2B). In contrast, beta diversity analysis revealed a significant difference in microbial community structure between the two groups. A PCoA based on Bray–Curtis dissimilarity easily separated the samples into two distinct groups corresponding to the CMS2/3 and CMS4 cohorts (Figure 2C). The dissimilarity in microbial community between the two clusters was statistically significant (PERMANOVA, F = 1.9, *p* = 0.034), suggesting that the CMS4 subtype possesses a distinct microbial community.

A random forest model was constructed to classify CMS groups based on microbial community profiles from the TCGA-COAD dataset. Our model was able to distinguish between CMS2/3 and CMS4 subtypes based on microbial composition, achieving an estimated accuracy of 63.4% (out-of-bag error rate: 36.6%), which is significantly greater than random chance (50%). This demonstrates that distinct microbial signatures that differentiate these two molecular subtypes are present and can be captured by our random forest model. Given the model’s ability to classify the groups, we next sought to identify the specific bacterial taxa that were most influential in driving this classification. We therefore analyzed their contribution to the Gini importance (mean decrease in Gini) in our model (Appendix A). This analysis identified several gut bacteria known to be associated with CRC as significant predictors (Figure 2D). Furthermore, group-specific importance scores of each microbe’s contribution to classifying a specific group revealed that *Bacteroides* made a greater contribution to identifying the CMS4 group, while *Fusobacterium* contributed more to identifying the CMS2/3 group (Figure 2E).

### 3.3. Bacteroides fragilis Enhances Growth and Changes the Gene Expression Profile into CMS4 in Co-Cultured Patient-Derived Organoids

To assess the functional role of ETBF in modulating molecular subtype identity in colorectal cancer, we used PDOs harboring mutations in *APC*, *KRAS*, and *TP53*, which were classified as CMS2 under standard organoid culture conditions but originated from a CMS4 CRC tissue (Appendix A). First, we confirmed bacterial attachment to PDOs, using ETBF labeled with a fluorescence tracker (Appendix A).

To mimic the TME, PDOs were co-cultured with stromal fibroblasts (CCD-18Co) and immune cells (THP-1) in a transwell plate. This co-culture system was then exposed to ETBF under two experimental conditions: condition 1 involved ETBF exposure restricted to the co-cultured 18Co/THP-1 cells in the lower chamber (PDO, 18Co/THP-1+ETBF), and condition 2 involved ETBF treatment of both the PDOs in the upper chamber and the 18Co/THP-1 cells in the lower chamber (PDO+ETBF, 18Co/THP-1+ETBF) (Figure 3A). After 7 days, organoid growth was assessed by size distribution (Figure 3B). A significant increase in the proportion of large organoids (>200 µm) was observed under both ETBF-treated conditions compared with untreated controls (*p* < 0.05). Notably, organoids in condition 2 exhibited a greater increase in size relative to condition 1 although the difference was not statistically significant (Figure 3C).

Next, we conducted transcriptome sequencing to analyze the gene expression landscape of CMSs under each co-culture condition. Our analysis revealed shifts in CMS classification dependent on the co-culture conditions (Figure 3D). In condition 2, which exposed both PDOs and co-cultured stromal and immune cells to ETBF, the PDOs shifted transcriptomically from a CMS2-like to a CMS4-like gene expression profile. In contrast, while the control group maintained its original CMS2 classification, PDOs from condition 1, in which only stromal/immune cells were treated with ETBF, transitioned to a non-consensus subtype.

To investigate the enrichment of CMS4-specific molecular characteristics between conditions, GSVA was performed using the CMS4 gene set as defined by CMScaller (Figure 3E). We then compared GSVA scores across the control, condition 1 (ETBF treatment to 18Co/THP-1 co-cultured cells only), and condition 2 (ETBF treatment to both PDOs and 18Co/THP-1 co-cultured cells) groups. The control group exhibited the fewest CMS4 molecular characteristics, indicated by a GSVA score of −0.5124. In condition 1, the suppressed CMS4 molecular characteristics were partially restored in the organoids compared to the control, with a GSVA score of −0.0769. In particular, the condition 2 group displayed the strongest CMS4 molecular signature, with an average GSVA score of 0.3092. These results collectively demonstrate that the presence of ETBF, particularly when interacting with both PDOs and the tumor microenvironment components, markedly enhances CMS4 molecular characteristics of tumor cells, suggesting subtype reprogramming and tumor growth promotion. To further elucidate the impact of ETBF on CMS4-associated transcriptional programs, we analyzed gene expression changes specifically within the CMS4 gene set. We compared ETBF-treated PDO/18Co/THP-1 co-cultures to control (PDO/18Co/THP-1 only) conditions and identified 110 CMS4 genes that were robustly upregulated upon ETBF exposure (Appendix A). These upregulated genes are predominantly involved in mesenchymal transition, extracellular matrix remodeling, and stromal activation, key features of the CMS4 subtype.

### 3.4. ETBF-Treated Co-Cultured PDOs Acquire a Transcriptional Profile Similar to CMS4 Colon Tumors

To investigate the degree of transcriptional resemblance of ETBF-treated co-cultured PDOs and the original CMS4 CRC tissue, we performed a Spearman correlation analysis using RNA-seq data from five experimental conditions: CMS4 CRC tissue, matched PDOs, and PDOs with or without ETBF and/or 18Co/THP-1 co-culture (conditions 1 and 2 in Figure 3). As shown in Figure 4A, the PDOs originated from CMS4 CRC were moderately correlated with the tumor tissue (r = 0.77), and a strong correlation (r = 0.80) was observed between the PDOs from condition 2 and the original CMS4 CRC tissue. In contrast to condition 2, lower correlation values with the CMS4 CRC tissue were observed in the controls (r = 0.71) and in condition 1 (r = 0.69). These results suggest that both ETBF exposure and TME context are critical for re-establishing the original CMS4 gene expression program in vitro, indicating the reprogramming potential of ETBF within a tumor microenvironment.

In addition, KEGG pathway enrichment analysis comparing ETBF-treated, 18Co/THP-1 co-cultured PDOs with untreated co-cultured controls revealed upregulation of significant pathways (Figure 4B). Among the top 10 enriched pathways, cytokine–cytokine-receptor interactions exhibited the highest gene ratio and a highly significant adjusted *p*-value (<0.001), indicating a central role. Other upregulated pathways include those for focal adhesion, cytoskeleton in muscle cells, and NOD-like receptor signaling, emphasizing their involvement in cell signaling and microenvironmental communication. Additionally, pathways such as ECM-receptor interaction and bacterial invasion of epithelial cells were enriched, although at lower gene ratios (Figure 4B, top). Similarly, KEGG analysis of patient CRC tissue compared with its corresponding PDOs demonstrated strong enrichment of cytokine–cytokine-receptor interactions (FDR-adjusted *p*-value < 0.001), ECM-receptor interaction, and PI3K-Akt signaling pathway (Figure 4B, bottom).

These shared pathway signatures suggest a convergence in molecular activation states between ETBF-treated, 18Co/THP-1 co-cultured PDOs and native tumor tissues, indicating potential common mechanisms by ETBF in TME-mediated CRC progression.

To further investigate whether these shared molecular features include aggressive tumor behaviors, we examined the enrichment of migration and invasion gene sets at the transcriptomic level. GSEA using a migration/invasion super-set, constructed from various migrated/invasion gene sets of MSigDB, revealed significant enrichment of these programs in ETBF-treated PDO/18Co/THP-1 co-cultures compared with controls. Furthermore, heatmap analysis of super-set that overlap with the CMS4 gene set showed coordinated upregulation in ETBF-treated samples (Appendix A). These results indicate that ETBF not only induces global molecular convergence with CMS4 tumors but also specifically promotes migration- and invasion-associated gene expression, highlighting its role in driving aggressive tumor phenotypes.

## 4. Discussion

Colorectal cancer remains prevalent worldwide, and optimal treatment is complicated by significant intra- and inter-tumoral heterogeneity. Among molecular subtypes of CRC, the CMS4 subtype is associated with poor prognoses and a high likelihood of metastasis. Although there have been new developments of tumor PDO models for drug discovery and precision medicine, in vitro modeling of CMS4 tumors remains limited due to its strong dependence on TME components [4,33]. Previous studies have reported that the unique gene signatures of CMS4 are often lost under standard PDO culturing, which lacks key TME components such as fibroblasts [4,34,35,36]. Our study directly addresses this major limitation by demonstrating that *B. fragilis*, a gut bacterium enriched in CMS4 tumors, can drive a shift from the epithelial CMS2 to the mesenchymal CMS4 subtype when co-cultured with TME components.

Crucially, our TME-mimicking co-culture model successfully recapitulated this phenotypic conversion. PDOs exposed to ETBF exhibited a transcriptomic shift from CMS2 towards a CMS4-like profile and a significant increase in size only when co-cultured with TME cells (fibroblasts and immune cells), unlike PDOs co-cultured with TME cells alone, or ETBF applied only to TME cells. Significantly, the transcriptome profile of these converted PDOs showed a strong correlation (r = 0.80) with that of the original patient’s CMS4 tumor tissue, validating that our model effectively mimics the in vivo state. This suggests that a complex interplay of PDOs with the bacteria and TME components is essential for inducing and maintaining the CMS4 phenotype of tumor cells.

Our initial identification of *B. fragilis* as a key species in our cohort was strongly validated by a separate analysis of the large TCGA-COAD dataset. 16S rRNA sequencing revealed increased colonization based on DNA abundance, while the random forest model identified *B. fragilis* as a top CRC-associated discriminator based on its transcriptional levels in the TCGA transcriptomic data. This concordance between two independent cohorts using distinct molecular measures provides compelling evidence that the significant role of *B. fragilis* is likely a generalizable feature of CMS4 CRC, rather than a cohort-specific artifact.

The mechanisms underlying this potential subtype shift may involve the well-characterized pathogenic activity of ETBF. Previous studies have established that ETBF promotes CRC progression through secretion of *B. fragilis* toxin (*BFT*), which cleaves E-cadherin, disrupts epithelial integrity, and activates the β-catenin and NF-kB signaling pathways [24,37,38]. Consistent with these reports, our KEGG pathway analysis revealed significant upregulation of cytokine–cytokine-receptor interaction and ECM-receptor interactions in ETBF-treated co-cultures (Figure 4B). This suggests that ETBF may trigger an inflammatory cascade and extracellular matrix (ECM) remodeling similar to that observed in CMS4 tumors, although specific signaling intermediates were not directly assessed in this study.

While our functional assays were conducted using a focused patient-derived model, the observed phenotypic shift towards CMS4 aligns with previous studies, validating the robustness of our model. For instance, other groups have independently demonstrated that ETBF promotes epithelial–mesenchymal transition (EMT) and metastasis [39]. Furthermore, the induction of stemness markers and inflammatory cytokines in our transcriptomic analysis recapitulates patterns reported in other ETBF-associated CRC models [22]. Importantly, our study extends these findings by showing that ETBF promotes the transition to the aggressive CMS4 subtype within the TME, a connection not previously established.

These findings suggest a clinically relevant possibility of dynamic CMS switching in vivo, where tumor–microbiome interactions could drive the transition from CMS2 to CMS4. If such switching occurs under therapeutic pressure, tumors may escape treatment by acquiring the invasive traits typical of CMS4, ultimately increasing the risk of recurrence. In this context, the co-culture model established here provides a tractable system for further dissecting the biological basis of this switching and for identifying potential intervention strategies.

This study has several experimental limitations that should be addressed in future research. First, the lack of matched normal mucosa in our 16S rRNA analysis limits our ability to distinguish tumor-specific enrichment from general dysbiosis, although our primary aim was to delineate heterogeneity within CRC subtypes. Second, while our TCGA analysis provides independent validation, it is limited to the genus level, whereas our primary 16s rRNA sequencing allowed for the identification of *B. fragilis*. Third, the current PDO model relied on a single ETBF strain and simplified TME components relative to in vivo conditions. Therefore, further validation using animal models will be required to confirm the relevance of these findings in a physiological context. Most notably, the absence of non-toxigenic *B. fragilis* (NTBF) or BFT-deficient mutant controls prevents us from definitively attributing the observed CMS4 transition exclusively to BFT.

In addition, the relatively small number of patient-derived samples represents a limitation. To increase the generalizability of our results, we conducted an additional analysis using the large, independent TCGA-COAD cohort, which yielded consistent conclusions and confirmed the robustness of our findings. In clinical data, however, potential confounding factors, such as dietary habits and recent antibiotic use, which can influence the tumor-associated microbiome, were not systematically controlled or recorded. Prospective studies that integrate these variables will be necessary to clarify the relationship between microbiome and CMS dynamics. Finally, the impact of ETBF colonization on the therapeutic response of CMS4 tumors remains unknown.

Nevertheless, the present data provides functional evidence that *B. fragilis* promotes CMS4-associated features and highlights the importance of integrating microbial–TME interactions in CRC research models.

## 5. Conclusions

We identified *B. fragilis* as a significantly enriched species in CMS4 CRC that is functionally associated with the mesenchymal phenotype. Using a TME-mimicking PDO co-culture model, we demonstrated that exposure to *B. fragilis* promotes a transcriptional shift toward CMS4, including the upregulation of its associated gene signatures. These findings suggest that *B. fragilis* may contribute to the acquisition of the aggressive CMS4 phenotype, highlighting the potential of targeting tumor–microbe–TME interactions as a novel therapeutic strategy for this aggressive CRC subtype.

## Figures and Tables

**Figure 1 cancers-17-03822-f001:**
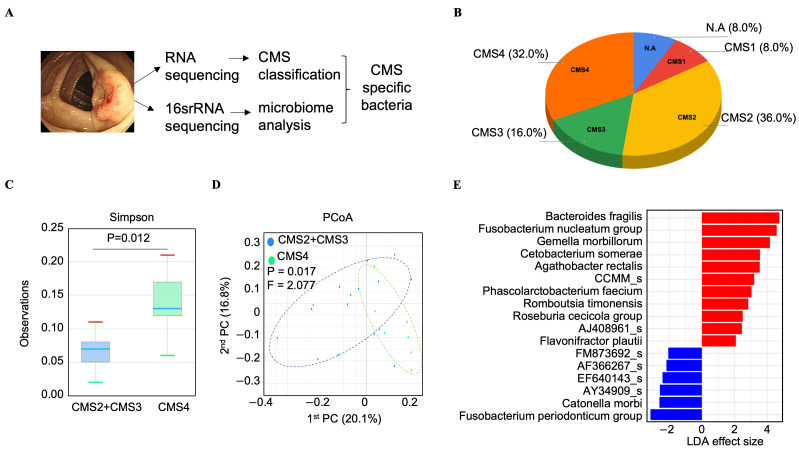
Bacterial species enriched in CMS2/3 and CMS4 tissue microbiomes. (**A**) Schematic outlining the process of obtaining tissues and the results of CMS specific bacteria. (**B**) The distribution of CMS of the 25 colorectal cancer (CRC) tumor tissues used in this study. (**C**) The alpha diversity index (Simpson) profile of CMS4 tissue microbiome and CMS2/3 tissue microbiomes. (**D**) Principal coordinates analysis (PCoA) showing percentage of variation in beta diversity and dissimilarity at species-level between CMS2/3 and CMS4 groups. (**E**) Linear discriminant analysis (LDA) effect size (LEfSe) plot of differentially abundant microbial species identified in the gut microbiome of CMS2/3 tissues, and CMS4 tissue with threshold > 2. Red bars denote genera with higher importance in CMS4, while blue bars indicate genera more important in CMS2/3. P: *p*-value; F: PERMANOVA F value; PC: principal coordinate.

**Figure 2 cancers-17-03822-f002:**
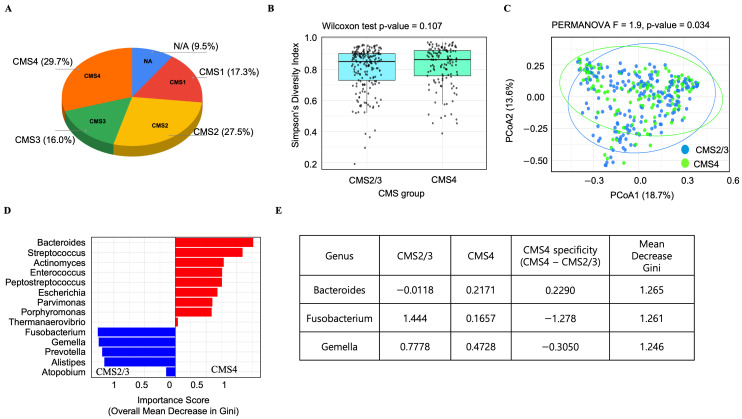
CMS-specific microbial signatures identified from TCGA-COAD dataset. (**A**) Distribution of 444 colorectal cancer (COAD) samples from the TCGA dataset according to their assigned CMS classification. (**B**) Alpha diversity index (Simpson) profile of the microbial communities, comparing CMS2/3 and CMS4 groups. Statistical significance was determined using the Wilcoxon rank-sum test. (**C**) Principal coordinates analysis (PCoA) based on Bray–Curtis dissimilarity at species-level, visualizing the beta diversity of microbial communities in CMS2/3 and CMS4 groups. The percentage of variation explained by each axis is shown. Group clustering significance was assessed by permutational multivariate analysis of variance (PERMANOVA). (**D**) Classification of colorectal cancer (CRC) molecular subtypes based on the associated microbiome using a random forest model (Appendix A depicts model performance). The plot specifically displays the feature importance of CRC-associated microbial genera derived from the model. Red bars denote genera with higher importance in CMS4, while blue bars indicate genera more important in CMS2/3. (**E**) CMS4 specificity of the top three genera with the highest feature importance scores. NA: Not applicable.

**Figure 3 cancers-17-03822-f003:**
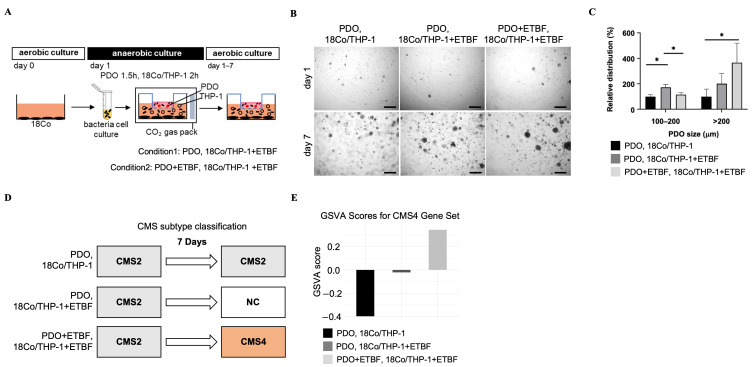
*Bacteroides fragilis* induces phenotypic and transcriptomic shifts toward CMS4 in co-cultured tumor organoids. (**A**) Experimental design of the 18Co/THP-1 co-culture condition and ETBF treatment. Patient-derived organoids (PDOs) were co-cultured with stromal (18Co) and immune (THP-1) cells in a trans-well system and treated with *B. fragilis* under two conditions: (Condition 1) bacteria applied to 18Co/THP-1 only (lower well); and (Condition 2) bacteria applied to both PDOs (upper well) and co-cultured 18Co/THP-1 cells (lower well). (**B**) Morphological analysis of PDOs following bacterial treatment in a co-culture condition. PDOs were co-cultured with TME components (18Co/THP-1 cells) and treated with *B. fragilis*. Representative bright-field images were acquired at day 1 and day 7 post-treatment. Scale bar, 500 µm. (**C**) After 7 days, PDO size distribution was quantified by sorting them into 100–200 µm and >200 µm categories. (**D**) CMS classification of PDOs after 7-day co-culture under control and *B. fragilis*-treated conditions. (**E**) Gene set variation analysis of 7-day co-cultured PDOs to assess enrichment of CMS4-related gene signatures. * *p*-value < 0.05.

**Figure 4 cancers-17-03822-f004:**
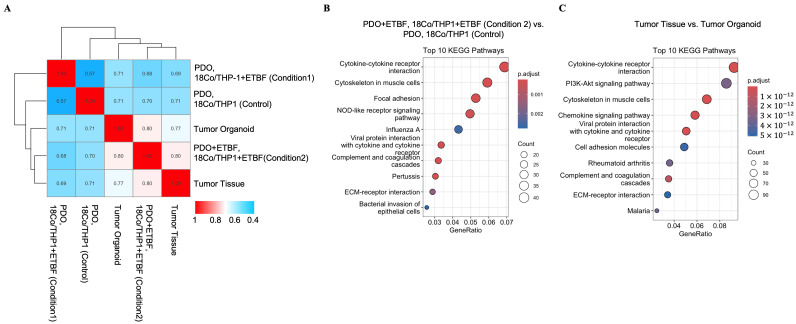
Functional and transcriptomic recapitulation of an original tumor tissue signature in ETBF-treated, 18Co/THP-1 co-cultured model. (**A**) Spearman correlation matrix heatmap comparing gene expression profiles of CMS-associated genes among five groups: untreated PDO co-cultured with ETBF-treated TME (18Co and THP-1 cells), untreated PDO and untreated 18Co/THP-1 (control), PDOs without co-culture and ETBF treatment, ETBF-treated PDO with treated 18Co/THP-1, and the original tumor tissue. Color scale represents the degree of correlation (r), with red indicating higher and blue indicating lower correlations. Hierarchical clustering was performed to group samples based on similarity. The matrix includes pairwise Spearman correlation coefficients for each condition. (**B**) Top 10 KEGG pathways significantly upregulated in PDOs co-cultured with stromal (18Co) and immune (THP-1) cells and treated with *Bacteroides fragilis*, compared with untreated co-cultured PDOs. (**C**) Top 10 KEGG pathways enriched in the original patient tumor tissue compared to the ETBF-untreated PDOs without co-culture, derived from that tumor. Enrichment analysis was performed using differentially expressed genes, with pathway significance assessed by FDR adjusted *p*-value thresholds.

**Table 1 cancers-17-03822-t001:** Patient cohort characteristics (*n* = number of patients).

Characteristics		N or Mean	% or Range
Age		64 years	26–92 years
Gender	Males	16	64
	Females	9	36
Biopsy Method	Colonoscopy	17	68
	Surgery	8	32
Site	Right	9	36
	Left	7	28
	Rectum	9	36
T stage	T1	2	8
	T2	5	20
	T3	14	56
	T4	4	16
N stage	N0	15	60
	N1	4	16
	N2	5	20
	unspecified	1	4
M stage	M0	20	80
	M1	2	8
	unspecified	3	12
AJCC Stage	0	1	4
	1	6	24
	2	7	28
	3	8	32
	4	3	12
Histology	Well Differentiated	8	32
	Moderately Differentiated	15	60
	Poorly Differentiated	2	8
MSI status	MSI-H	2	8
	MSI-L	0	0
	MSS	19	76
	N/A	4	16
KRAS mutation status	positive	8	32
	negative	13	52
	N/A	4	16
NRAS mutation status	positive	0	0
	negative	21	84
	N/A	4	16

## Data Availability

The RNA sequencing data generated in this study have been deposited in the ArrayExpress database at EMBL-EBI (https://www.ebi.ac.uk/arrayexpress (accessed on 5 September 2025) under accession number [E-MTAB-15616]. The 16S rRNA gene sequencing data have been deposited in the European Nucleotide Archive (ENA) under accession number [ERP179975]. Publicly available colorectal cancer datasets from The Cancer Genome Atlas (COAD and READ) were obtained through the Genomic Data Commons (GDC) portal (https://portal.gdc.cancer.gov/ (accessed on 5 September 2025). The results of RNA-seq processed and microbiome analysis, and custom R scripts used for analysis have been deposited in the GitHub (commit eb2b28c) repository ETBF_mesenchymal_subtype_CRC (latest) at https://github.com/kim-dk-ncc/ETBF_mesenchymal_subtype_CRC (accessed on 5 September 2025), which is openly available under the CC0 license.

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
