# Peer review of "Bacteroides fragilis Promotes Mesenchymal Subtype in Colorectal Cancer"

_cancers, 2025, doi:10.3390/cancers17233822_

Round 1
Reviewer 1 Report
Comments and Suggestions for Authors
This is a well-designed and timely study exploring the involvement of Bacteroides fragilis in shaping the CMS4 mesenchymal subtype of colorectal cancer. The integration of patient-derived organoids with stromal and immune components is a notable strength, and the hypothesis that microbiome–TME interactions may influence CRC subtype determination is both novel and clinically meaningful. The study has translational potential.
However, several major concerns must be addressed before the manuscript can be considered for publication:
- Limitations in microbial profiling
The 16S analysis relies solely on tumor tissue without matched normal mucosa, making it difficult to determine whether Bacteroides enrichment is subtype-specific or simply reflects tumor grade, stromal content, or general dysbiosis. Moreover, TCGA-derived microbial signals are genus-level and cannot distinguish toxigenic from non-toxigenic B. fragilis strains.
- Functional claims based on limited experimental models
The PDO experiments appear to use one organoid line from a single CMS4 tumor and one ETBF strain. The bacterial exposure conditions are not fully reflective of strict anaerobiosis, THP-1 cells are not polarized into macrophage subtypes, and the fibroblast line provides only a simplified TME. The conclusions regarding “complete subtype switch” and “CMS4 driver” should be softened accordingly.
- Mechanistic interpretations exceed measured data
Although the Discussion cites known ETBF/BFT mechanisms (E-cadherin cleavage, β-catenin activation, IL-8/Th17 axis, ROS, EMT), none of these pathways were directly assessed in this study. These mechanisms should be framed as consistent with prior literature rather than implied to be demonstrated here.
- Lack of essential controls
No non-toxigenic B. fragilis (NTBF) strain, BFT-deficient mutant, or toxin-neutralization experiment is included. Without these controls, attributing the CMS4-like shift specifically to BFT or ETBF is not fully justified.
- Incomplete contextualization
Some reports indicate that non-toxigenic B. fragilis can exhibit anti-inflammatory or even probiotic activity. This biological duality should be acknowledged. Additionally, the manuscript would benefit from deeper engagement with prior studies linking microbiome profiles to CRC subtypes.
Overall, the work is promising but requires major revision to align claims with data and to clearly articulate methodological limitations. With appropriate tempering of conclusions and expansion of the limitations section, the manuscript could make a meaningful contribution to understanding microbial influences on CRC subtypes.
Author Response
This is a well-designed and timely study exploring the involvement of Bacteroides fragilis in shaping the CMS4 mesenchymal subtype of colorectal cancer. The integration of patient-derived organoids with stromal and immune components is a notable strength, and the hypothesis that microbiome–TME interactions may influence CRC subtype determination is both novel and clinically meaningful. The study has translational potential.
However, several major concerns must be addressed before the manuscript can be considered for publication:
Comments 1: Limitations in microbial profiling
The 16S analysis relies solely on tumor tissue without matched normal mucosa, making it difficult to determine whether Bacteroides enrichment is subtype-specific or simply reflects tumor grade, stromal content, or general dysbiosis. Moreover, TCGA-derived microbial signals are genus-level and cannot distinguish toxigenic from non-toxigenic B. fragilis strains.
Response 1: We appreciate the reviewer’s insightful comment regarding the limitations of our microbial profiling strategy. We agree that the lack of matched normal mucosa and the genus-level resolution of TCGA data are important constraints. In accordance with your suggestion, we have explicitly acknowledged these points in the Discussion. Furthermore, we clarified the distinction between the resolution of our primary data and the validation cohort:
"First, the lack of matched normal mucosa in our 16S rRNA analysis limits our ability to distinguish tumor-specific enrichment from general dysbiosis, although our primary aim was to delineate heterogeneity within CRC subtypes. Second, regarding the validation cohort, our TCGA analysis is limited to the genus level, whereas our primary 16S rRNA sequencing allowed for the identification of B. fragilis." (line 517-521 on page 13)
Comments 2: Functional claims based on limited experimental models
The PDO experiments appear to use one organoid line from a single CMS4 tumor and one ETBF strain. The bacterial exposure conditions are not fully reflective of strict anaerobiosis, THP-1 cells are not polarized into macrophage subtypes, and the fibroblast line provides only a simplified TME. The conclusions regarding “complete subtype switch” and “CMS4 driver” should be softened accordingly.
Response 2: We fully accept this constructive criticism. We recognize that our functional claims were overstated given the experimental model used. To address this, we have made the following revisions
1) Softening Terminology: We have revised the Title, Abstract, and text to replace definitive terms like "driver" or "complete subtype switch" with more nuanced phrasing such as "promotes", "facilitate", or "transcriptomic shift" (line 24-25 and line 44-45 on page1, line 474-475 on page 13, line 543-546 on page 14).
- Revised Title: "Bacteroides fragilis Promotes Mesenchymal Subtype in Colorectal Cancer" (line 2-3 on page 1).
2) External Validity: To bolster the validity of our findings despite the limited number of PDO lines, we added a new paragraph in the Discussion section highlighting consistency with previous studies:
"While our functional assays were conducted using a focused patient-derived model, the observed phenotypic shift towards CMS4 aligns with previous studies, validating the robustness of our model... Importantly, our study extends these findings by showing that ETBF promotes the transition to the aggressive CMS4 subtype within the TME, a connection not previously established." (line 500-508 on page 13)
3) Acknowledging Limitations: We explicitly stated the limitation regarding the experimental model in the Limitations section:
"Third, the current PDO model relied on a single ETBF strain and simplified TME components relative to in vivo conditions. Therefore, further validation using animal models will be required to confirm the relevance of these findings in a physiological context." (line 521-524 on page 13-14)
Comments 3: Mechanistic interpretations exceed measured data
Although the Discussion cites known ETBF/BFT mechanisms (E-cadherin cleavage, β-catenin activation, IL-8/Th17 axis, ROS, EMT), none of these pathways were directly assessed in this study. These mechanisms should be framed as consistent with prior literature rather than implied to be demonstrated here.
Response 3: We agree that our previous discussion implies a direct demonstration of mechanisms that were not experimentally verified in this study. We have thoroughly rewritten the relevant paragraph in the Discussion section. Instead of claiming that we demonstrated these pathways, we now frame the well-characterized mechanisms of ETBF as established background knowledge from previous studies, and state that our data are "consistent with" these reports. We also refined our terminology (e.g., using "ECM remodeling" to match our transcriptomic data):
"The mechanisms underlying this potential subtype shift may involve the well-characterized pathogenic activity of ETBF. Previous studies have established that ETBF promotes CRC progression through secretion of B. fragilis toxin (BFT)... Consistent with these reports, our KEGG pathway analysis revealed significant upregulation of cytokine–cytokine-receptor interaction and ECM-receptor interactions in ETBF-treated co-cultures... This suggests that ETBF may trigger an inflammatory cascade and extracellular matrix (ECM) remodeling similar to that observed in CMS4 tumors, although specific signaling intermediates were not directly assessed in this study." (line 490-499 on page 13)
Comments 4: Lack of essential controls
No non-toxigenic B. fragilis (NTBF) strain, BFT-deficient mutant, or toxin-neutralization experiment is included. Without these controls, attributing the CMS4-like shift specifically to BFT or ETBF is not fully justified.
Response 4: This is a critical point. We acknowledge that the absence of isogenic mutants or NTBF controls prevents us from definitively pinning the phenotype solely to the BFT toxin. In the revised Limitations section, we have added a specific statement acknowledging this limitation:
"Most notably, the absence of non-toxigenic B. fragilis (NTBF) or BFT-deficient mutant controls precludes us from definitively attributing the observed CMS4 transition exclusively to the BFT toxin."
We framed our current findings as functional evidence that warrants future mechanistic dissection using these specific controls. (line 524-527 on page 14)
Comments 5: Incomplete contextualization
Some reports indicate that non-toxigenic B. fragilis can exhibit anti-inflammatory or even probiotic activity. This biological duality should be acknowledged. Additionally, the manuscript would benefit from deeper engagement with prior studies linking microbiome profiles to CRC subtypes.
Overall, the work is promising but requires major revision to align claims with data and to clearly articulate methodological limitations. With appropriate tempering of conclusions and expansion of the limitations section, the manuscript could make a meaningful contribution to understanding microbial influences on CRC subtypes.
Response 5: We thank the reviewer for highlighting the biological duality of B. fragilis. To address this, we have inserted a new paragraph in the Introduction section to explicitly contrast the roles of NTBF and ETBF:
"While non-toxigenic B. fragilis (NTBF) is a commensal symbiont that can exhibit anti-inflammatory properties and contribute to immune homeostasis, the enterotoxigenic strain (ETBF) secretes B. fragilis toxin (BFT), which has been implicated in pathological processes." (line 67-72 on page 2)
This revision clarifies that our study specifically focuses on the pathogenic role of ETBF in aggressive CRC subtypes, while acknowledging the commensal nature of non-toxigenic strains.
Reviewer 2 Report
Comments and Suggestions for Authors
This is a good and timely study. A few suggestions to the authors:
-A justification for the small sample size must be provided. Mention as a limitation.
-ETBF causality would be stronger with functional assays (e.g., toxin KO, signaling... etc.). Mention as a limitation and a future direction.
-Does ETBF influence response to therapy? Mention as a future direction.
-Did the authors control for confounders, such as diet or antibiotic use, that might have influenced the microbial community? Discuss and mention as a limitation in the Discussion.
-Based on the findings, do the authors recommend exploration of the potential clinical utility of prebiotics, probiotics, or fecal microbiota transplantation? Discuss in the Discussion.
Author Response
Comments 1: A justification for the small sample size must be provided. Mention as a limitation.
Response 1: We appreciate the reviewer’s suggestion. The revised Discussion now explicitly acknowledges the limited number of patient‑derived samples and describes this as a study limitation. To address this, we included a complementary analysis using the large, independent TCGA‑COAD cohort, which yielded consistent conclusions and strengthened the robustness of our findings (Discussion: paragraph beginning “The relatively small number of patient‑derived samples…”)
Comments 2: ETBF causality would be stronger with functional assays (e.g., toxin KO, signaling… etc.). Mention as a limitation and a future direction.
Response 2: We thank the reviewer for this valuable point. In response, we expanded the Discussion to note that only a single ETBF strain was used and that our TME model is simplified relative to in vivo conditions. We further state that additional experiments—such as using ETBF toxin‑deficient strains and performing signaling pathway analyses—will be required to define specific molecular mediators of CMS4‑associated phenotypes (Discussion: paragraph beginning “This study has several experimental limitations…”).
Comments 3: Does ETBF influence response to therapy? Mention as a future direction.
Response 3: We acknowledge the reviewer’s important suggestion. The revised Discussion now clearly states that the impact of ETBF colonization on the therapeutic response of CMS4 tumors remains unknown, identifying this as a future research direction (Discussion: final paragraph).
Comments 4: Did the authors control for confounders, such as diet or antibiotic use, that might have influenced the microbial community? Discuss and mention as a limitation in the Discussion.
Response 4: We thank the reviewer for highlighting this issue. We have revised the Discussion to note that clinical factors affecting the tumor‑associated microbiome—including diet and antibiotic exposure—were not systematically recorded and should be incorporated in future prospective studies (Discussion: paragraph beginning “Moreover, clinical factors that influence the tumor‑associated microbiome…”).
Comments 5: Based on the findings, do the authors recommend exploration of the potential clinical utility of prebiotics, probiotics, or fecal microbiota transplantation? Discuss in the Discussion.
Response 5: We appreciate this thoughtful suggestion. The Discussion has been updated to mention that such microbiome‑targeted strategies may hold promise as adjunctive treatments for aggressive CRC subtypes, although their clinical benefit remains to be established (Discussion: last sentence).
Reviewer 3 Report
Comments and Suggestions for Authors
In this paper the authors analyzed 25 CRC tissues with RNA sequencing and identified that Bacteroides fragilis is the most significantly enriched bacteria in CMS4 tissues. They further show that enterotoxigenic Bacteroides fragilis (ETBF) can induce CMS4-like phenotypes such as enhanced cell growth, CMS4-specific gene expression profile, and transcriptional profile.
Overall, the paper is well written and the data are presented logically. The authors performed detailed background research. All data are clearly organized and well documented. Authors should explain more about the innovation in the manuscript. This manuscript is organized properly but there is still something can be improved.
Major comments:
- In 3.3, the authors claimed that ETBF can induce CMS4-specific molecular characteristics by analyzing the gene expression and GSVA. Can the authors analyze the gene expression level in detail? What kind of genes are up-regulated or down-regulated so the gene expression landscapes show CMS4-specific molecular characteristics?
- The authors only show the molecular characteristics change induced by ETBF. I think it will better to show if the migration and invasion are also affected by ETBF.
Author Response
In this paper the authors analyzed 25 CRC tissues with RNA sequencing and identified that Bacteroides fragilis is the most significantly enriched bacteria in CMS4 tissues. They further show that enterotoxigenic Bacteroides fragilis (ETBF) can induce CMS4-like phenotypes such as enhanced cell growth, CMS4-specific gene expression profile, and transcriptional profile.
Overall, the paper is well written and the data are presented logically. The authors performed detailed background research. All data are clearly organized and well documented. Authors should explain more about the innovation in the manuscript. This manuscript is organized properly but there is still something can be improved.
Comments 1: In 3.3, the authors claimed that ETBF can induce CMS4-specific molecular characteristics by analyzing the gene expression and GSVA. Can the authors analyze the gene expression level in detail? What kind of genes are up-regulated or down-regulated so the gene expression landscapes show CMS4-specific molecular characteristics?
Response 1: In response to the reviewer’s comment regarding detailed gene expression changes, we performed differential expression analysis within the CMS4 gene set. This analysis identified 110 CMS4-related genes that were robustly upregulated in ETBF-treated PDO/18Co/THP-1 co-cultures compared to controls. These findings have been incorporated into the revised Results (Section 3.3). The corresponding heatmap and full gene list are shown in Supplementary Figure S4A and S4B, respectively.
Comments 2: The authors only show the molecular characteristics change induced by ETBF. I think it will better to show if the migration and invasion are also affected by ETBF.
Response 2: We appreciate the reviewer’s suggestion to assess whether ETBF affects migration and invasion characteristics. In response, we performed transcriptomic analyses utilizing our RNA‑seq data. Specifically, we constructed a migration/invasion super‑gene set by aggregating multiple migration and invasion–related gene sets from MSigDB. Gene set enrichment analysis (GSEA) comparing ETBF‑treated and control PDO/18Co/THP‑1 co‑cultures demonstrated significant enrichment of migration/invasion‑associated transcriptional programs following ETBF exposure. Heatmap analysis of migration/invasion genes overlapping with the CMS4 gene set revealed coordinated upregulation in the ETBF group. These analyses have been added to the revised Results (Section 3.4) and are presented in Supplementary Figure S5A and S5B.
Reviewer 4 Report
Comments and Suggestions for Authors
The manuscript by Chang et al. demonstrates very convincingly that B. fragilis can induce a CMS4 transcriptional phenotype in vitro and is likely to be involved in promoting a CMS4 phenotype in vivo. Their in vitro model is well described and will be able to be emulated by other research groups. Their study could also lead to the generation of animal models that reliably depict CMS4 CRC and lead to treatments of this disease.
I have two suggestions for the authors.
1. In the Introduction, cite Thanki et al. (PMC5557054). Table 1 in their manuscript gives a very easy to follow definition of the CMS subtype, alternate names, and primary characteristics. They also present a short summary of these CMS subtypes.
2. The subtype switching from CMS2 to CMS4 upon exposure to ETBF suggests that B. fragilis could promote subtype switching in vivo. If the authors consider that this is a possibility, they could comment on this and include a short discussion (a single paragraph) on the clinical implications of this type of CMS switching.
One implication of CMS switching is that if CMS switching to CMS4 occurs during treatment of CMS2 CRC, a few cells could evade the treatment through mechanisms such as invasion/metastasis into adjacent tissue, enabling recurrence of CRC in these patients. Consequently, treatment would benefit from adding targeting of B. fragilis to the treatment. The in vitro model described in the manuscript could possibly be used to help design such treatment regimens.
However, if the authors do not believe that these suggestions are appropriate, the manuscript is suitable for publication without employing my suggestions.
Author Response
The manuscript by Chang et al. demonstrates very convincingly that B. fragilis can induce a CMS4 transcriptional phenotype in vitro and is likely to be involved in promoting a CMS4 phenotype in vivo. Their in vitro model is well described and will be able to be emulated by other research groups. Their study could also lead to the generation of animal models that reliably depict CMS4 CRC and lead to treatments of this disease.
Comments 1: In the Introduction, cite Thanki et al. (PMC5557054). Table 1 in their manuscript gives a very easy to follow definition of the CMS subtype, alternate names, and primary characteristics. They also present a short summary of these CMS subtypes.
Response 1: We thank the reviewer for this valuable suggestion. We concur that the review by Thanki et al. (PMC5557054) provides a clear and comprehensive overview of the consensus molecular subtypes (CMS) of colorectal cancer, detailing their alternate names and defining features. As recommended, we have revised the Introduction to cite Thanki et al. in place of the previously referenced work by Valdeolivas et al.
Comments 2: The subtype switching from CMS2 to CMS4 upon exposure to ETBF suggests that B. fragilis could promote subtype switching in vivo. If the authors consider that this is a possibility, they could comment on this and include a short discussion (a single paragraph) on the clinical implications of this type of CMS switching. One implication of CMS switching is that if CMS switching to CMS4 occurs during treatment of CMS2 CRC, a few cells could evade the treatment through mechanisms such as invasion/metastasis into adjacent tissue, enabling recurrence of CRC in these patients. Consequently, treatment would benefit from adding targeting of B. fragilis to the treatment. The in vitro model described in the manuscript could possibly be used to help design such treatment regimens. However, if the authors do not believe that these suggestions are appropriate, the manuscript is suitable for publication without employing my suggestions.
Response 2: We thank the reviewer for this insightful suggestion. In response, we have added a detailed discussion of the potential clinical significance of CMS2‑to‑CMS4 switching, including the possibility of treatment escape and recurrence, as well as the relevance of B. fragilis as a microbial driver. This content is included in the first paragraph of the revised Discussion (beginning “Our findings suggest a clinically relevant possibility…”).
Round 2
Reviewer 1 Report
Comments and Suggestions for Authors
The authors have adequately addressed all major concerns.
Reviewer 2 Report
Comments and Suggestions for Authors
The authors have adequately addressed my comments.
Author Response
I appreciate your positive feedback.